# Brightness modulations of our nearest terrestrial planet Venus reveal atmospheric super-rotation rather than surface features

Y. J. Lee [1✉], A. García Muñoz[1], T. Imamura[2], M. Yamada [3], T. Satoh [4], A. Yamazaki[4,5] & S. Watanabe[6]

Terrestrial exoplanets orbiting within or near their host stars' habitable zone are potentially apt for life. It has been proposed that time-series measurements of reflected starlight from such planets will reveal their rotational period, main surface features and some atmospheric information. From imagery obtained with the Akatsuki spacecraft, here we show that Venus' brightness at 283, 365, and 2020 nm is modulated by one or both of two periods of 3.7 and 4.6 days, and typical amplitudes <10% but occasional events of 20–40%. The modulations are unrelated to the solid-body rotation; they are caused by planetary-scale waves superimposed on the super-rotating winds. Here we propose that two modulation periods whose ratio of large-to-small values is not an integer number imply the existence of an atmosphere if detected at an exoplanet, but it remains ambiguous whether the atmosphere is optically thin or thick, as for Earth or Venus respectively. Multi-wavelength and long temporal baseline observations may be required to decide between these scenarios. Ultimately, Venus represents a false positive for interpretations of brightness modulations of terrestrial exoplanets in terms of surface features.

[1] Technische Universität Berlin, Berlin, Germany. [2] GSFS, Univ. of Tokyo, Kashiwa, Japan. [3] Planetary Exploration Research Center (PERC), Narashino, Japan. [4] Institute of Space and Astronautical Science (ISAS/JAXA), Sagamihara, Japan. [5] Graduate School of Science, Univ. of Tokyo, Tokyo, Japan. [6] Hokkaido Information University, Ebetsu, Japan. ✉email: y.j.lee@astro.physik.tu-berlin.de

As the search for terrestrial exoplanets advances, and the technology that will enable their characterization matures, it becomes important to establish observational diagnostics that inform us about their surfaces and atmospheres, and testing such diagnostics over a variety of conditions. Time-series measurements of a planet's reflected starlight potentially provide an avenue to map a planet's surface as photometric variability and surface inhomogeneity are interconnected.

The idea has been extensively developed[1–7], and convincingly demonstrated for Earth with the Deep Space Climate Observatory (DSCOVR) space-based photometry gathered for more than 2 years and 10 wavelengths from 320 to 780 nm[6]. The idea is valid if the atmosphere is optically thin (as Earth's), so that stellar photons reach the surface and escape back to space. In general, it may be non-trivial to determine if a small-mass exoplanet has an atmosphere, let alone if it is thin or thick. Clouds, if present, will interfere with the surface signal and introduce additional temporal variability[8], but long-term exposures may filter out such effects[1,6].

A key characteristic of the Earth's brightness modulation is that its periodogram shows a dominant peak at $P = 1$ day and at the fractional periods $\frac{1}{2}$, $\frac{1}{3}$ and $\frac{1}{4}$ days for all wavelengths in the DSCOVR dataset[6]. The 1-day signal originates from the Earth's rotation, whereas the shorter-period signals are related to the details in the distribution of continents and oceans[2,6].

Venus is currently outside the so-called habitable zone (HZ, the circumsolar region within which liquid water might occur at the planet's surface) but it was possibly habitable in the past[9]. Venus' equilibrium temperature is $T_{eq} = 230$ K, not much different from Earth's $T_{eq} = 254$ K. A huge greenhouse effect however keeps Venus' surface temperature at 735 K, too hot to allow for liquid water. It remains unclear at what point Venus drifted into that state if, as is usually thought, both planets might have had similar conditions in their early days[10,11].

Exo-Venuses, i.e. planets near the inner boundary of their host stars' HZ, are expected to be abundant[9,12,13]. Thus, it is important to devise diagnostics beyond first-order factors such as the orbital distance that will help distinguish between genuine exo-Earths, with mild temperatures suitable for life, and exo-Venuses. The question is complex yet key in the characterization of terrestrial exoplanets and will require multiple approaches to address it.

Here, we show the photometric time series of Venus in anticipation of what might be expected for exo-Venus observations. We find two distinct periods (3.7 and 4.6 days) in the modulation of the reflected sunlight. These periods are ~60 times shorter than Venus' solid-body rotational period, and so they are unrelated to the surface. Rather, they originate from the super-rotating background winds and superimposed planetary-scale waves. We show their wavelength dependence at 283, 365, and 2020 nm, and their temporal variability in Venus' disk-integrated photometry data. We propose that the two distinct nearby peaks are a sign of the existence of an atmosphere, and that a long-baseline campaign of multi-wavelength observations (and the search for temporal variations in them) will help conclude that such an exoplanet might have a Venus-like thick atmosphere. Our study conveys the caution message that distinguishing between brightness modulations associated with the solid-body rotation of an exoplanet or its atmospheric winds will have to be carefully considered in future data analyses of exoplanets.

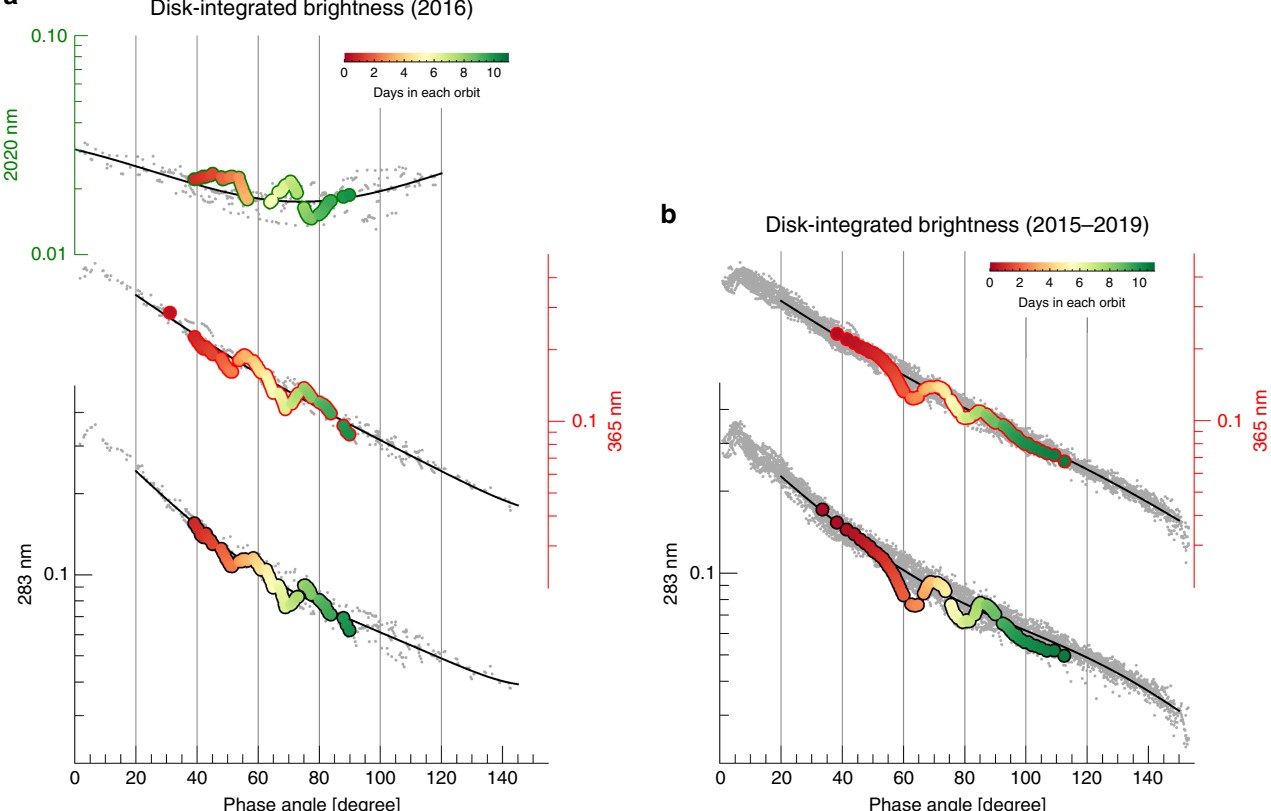

**Fig. 1 Venus phase curves over time.** The color bars indicate Earth days within one orbit (~11 days). The solid black lines are best-fitting 4-degree polynomials for each dataset, which we use as baselines to investigate temporal variations. **a** From top to bottom, brightness at 2020, 365, and 283 nm between March 31 and October 29, 2016 (gray dots). The largest temporal variation occurred on orbit 20 (color symbols) (See also Fig. 2). **b** UV brightness from Dec 2015 to Jan 2019 (gray dots). Unusually high brightness variations occurred on orbit 81 (color symbols).

## Results

**Multispectral photometry of Venus**. We investigated Venus in reflected sunlight using whole-disk imagery produced by the JAXA/Akatsuki spacecraft[14] in the ultraviolet (UV) and near infrared (NIR). The images were acquired with the UV camera (UVI) at effective wavelengths of 283 and 365 nm[15], and with the NIR camera (IR2) at an effective wavelength of 2020 nm[16]. Each set of 2–3 images was obtained within 9 min and thus the images in a set can be considered to be nearly simultaneous. The time interval between each set of images is typically 2 h, but can be up to a few days depending on the location of the spacecraft on its highly elliptical orbit.

As the whole disk of Venus can only be imaged from a distance sufficiently far from the planet, we used images taken before or after pericenter passage[14]. So while Akatsuki revolves around Venus every ~11 days, it obtains whole-disk images for ~10 days per orbit during the dayside-monitoring epoch. The sequence of observations alternates every 4 months between dayside- and nightside-monitoring epochs up to one Venusian year (225 days, ~8 months), when a new dayside–nightside sequence is started. For our analysis, we utilized a total of 5805 (283 nm) and 5840 (365 nm) UVI images obtained between 2015 and 2019, and 354 IR2 images obtained in 2016, at the end of which year the latter camera stopped working (see Methods, subsection "Image processing", and Supplementary Fig. 1 for details on the data).

We describe Venus' disk-integrated brightness in the usual form of a geometric albedo × phase law that depends on the Sun–Venus-spacecraft phase angle $\alpha$ but not on the planet's apparent size (see Eq. (4) in Methods, subsection "Image processing"). Hereafter, we refer to this size-normalized measure of brightness simply as the planet's brightness or phase-resolved albedo.

In the UV, the brightness generally decreases as $\alpha$ increases (Fig. 1)[17]. The abrupt variation at small phase angles is the glory, an optical phenomenon due to scattering from narrow-size distributions of cloud droplets[18–20]. The Venusian clouds are very thick (optical thickness $\tau \sim 30$ in the visible[21]), which prevents the access of the solar photons to the surface at UV-NIR wavelengths on the dayside. They also contain traces of an unknown absorber that produces the dark patterns seen in UV images (Fig. 2) and that absorbs most strongly at 350-380 nm[22,23]. Absorption by the unknown absorber and the $SO_2$ gas above the clouds reduce the brightness at 283 nm and result in a lower brightness at this wavelength[24]. Venus' main atmospheric gas, $CO_2$, absorbs strongly at 2020 nm[16,25] above the cloud top level at ~70 km, which reduces the NIR brightness to <0.03, an order of magnitude less than in the UV (Fig. 1). The NIR brightness increases for $\alpha > 80°$ due to scattering by the haze that exists above the clouds[26,27] and whose relative contribution against $CO_2$ absorption increases for high phase angles.

The phase-resolved albedo of Fig. 1 and the sequence of images of Fig. 2 demonstrate that Venus' brightness varies overtime at all three wavelengths. Modulations about the mean conditions are seen at each orbit and over most of the monitored phase angles, thus confirming their persistent nature. Orbits 20 and 81 (Fig. 1; (**a**) and (**b**) panels, respectively) exhibit particularly strong modulations with peak-to-peak amplitudes of ~20% in the UV and ~40% in the NIR (orbit 20) (see Methods, subsection "Mean phase curves and periodicity analysis", and Supplementary Figs. 3, 5). The amplitude of these modulations at the planetary scale was unknown to date.

The temporal variability in the phase-resolved albedo is clearly seen in our Supplementary Movies 1–2, and suggests multiple timescales associated with changes in the spatial distribution of absorbers and in the global cloud morphology. There is an anti-correlation between the UV and NIR brightness: an increase in

the UV is consistently accompanied by a decrease in the NIR and vice versa. This means that the main absorbers at the wavelengths investigated here (the unknown absorber and $SO_2$ in the UV; $CO_2$ in the NIR) affect the Venus brightness in opposite yet temporally related ways.

The disk-resolved images of Fig. 2 help understand the brightness modulations. Fig. 2a confirms that the UV–NIR brightness is indeed anti-correlated. Each column of images shows a nearly simultaneous snapshot of Venus at the three wavelengths. There is a peculiar, global scale synchronization between low and high latitudes; the NIR modulations occur at all latitudes, including middle and high latitudes (Fig. 2e), while the UV modulations occur mainly although not exclusively at low latitudes (Fig. 2f, g). This previously unknown behavior is likely related to the development of the known 'Y'-shape feature, which might result from a combination of Kelvin and Rossby atmospheric waves from low to high latitudes[28].

**Modulations of disk-integrated brightness**. At the cloud-top level probed by Akatsuki, the Venus atmosphere rotates in the same direction as the surface but 60–80 times faster[28], and thus it takes ~4–5 days for the zonal winds to circle the planet. This super-rotation occurs simultaneously with the vertical and horizontal oscillations at the cloud top level that drive the NIR and UV modulations in brightness, respectively.

To better characterize the temporal behavior of these modulations, we have defined brightness deviations with respect to a baseline constructed by fitting a 4th-order polynomial to the phase-resolved albedos (Fig. 1). The periodogram of these deviations (Fig. 3) reveals two distinct peaks at $P_1 \sim 3.7$ and $P_2 \sim 4.6$ days. Periods comparable to $P_1$ and $P_2$ have been reported before to describe the modulations in local brightness and wind velocities at Venus[29–32], and used to support their interpretation in terms of waves. The match between our periods and those reported elsewhere is particularly good when we focus separately on the low and middle latitudes, as has been customarily done in previous work (Supplementary Table 1 and Supplementary Fig. 8).

A period ~4 days was reported for the whole-disk brightness measurements at 365 nm made by the Pioneer Venus Orbiter spacecraft[33]. This is, however, the first instance that both periods are clearly identified in the disk-integrated brightness of Venus and at multiple wavelengths. Based on previous investigations at 365 nm[29–32], the $P_1$ and $P_2$ periods are associated with an equatorial Kelvin wave and a mid-latitude Rossby wave moving in the direction of the mean zonal flow at phase speeds somewhat faster and slower than the zonal winds, respectively.

Interestingly, the strengths and widths in the periodogram of the $P_1$- and $P_2$-period signals are very different at each wavelength. Both periods are confidently detected at 365 and 2020 nm, but only $P_1$ is noticeable at 283 nm. This suggests that the impact of each wave on the planet's brightness is affected by the latitudinal distribution of absorbers and their wavelength-dependence absorption properties (see Methods, subsection "The missing $P_2$ period at 283 nm in the context of Venus studies").

At low latitudes, both the unknown absorber and $SO_2$ gas are most abundant[34,35]. Their abundances as a function of altitude decrease rapidly upwards near the cloud top level[36,37]. The impact of $CO_2$ absorption in the NIR depends on slight changes in the cloud top altitudes[38]. The equatorial Kelvin wave causes vertical oscillations of all these absorbers at low latitudes[29], and consequently $P_1$ is apparent at all wavelengths. At mid-to-high latitudes, the 365 nm brightness shows strong latitudinal variations (dark spiral and bright polar hood[39]), while the 2020 nm brightness drops steeply towards high latitudes due to the

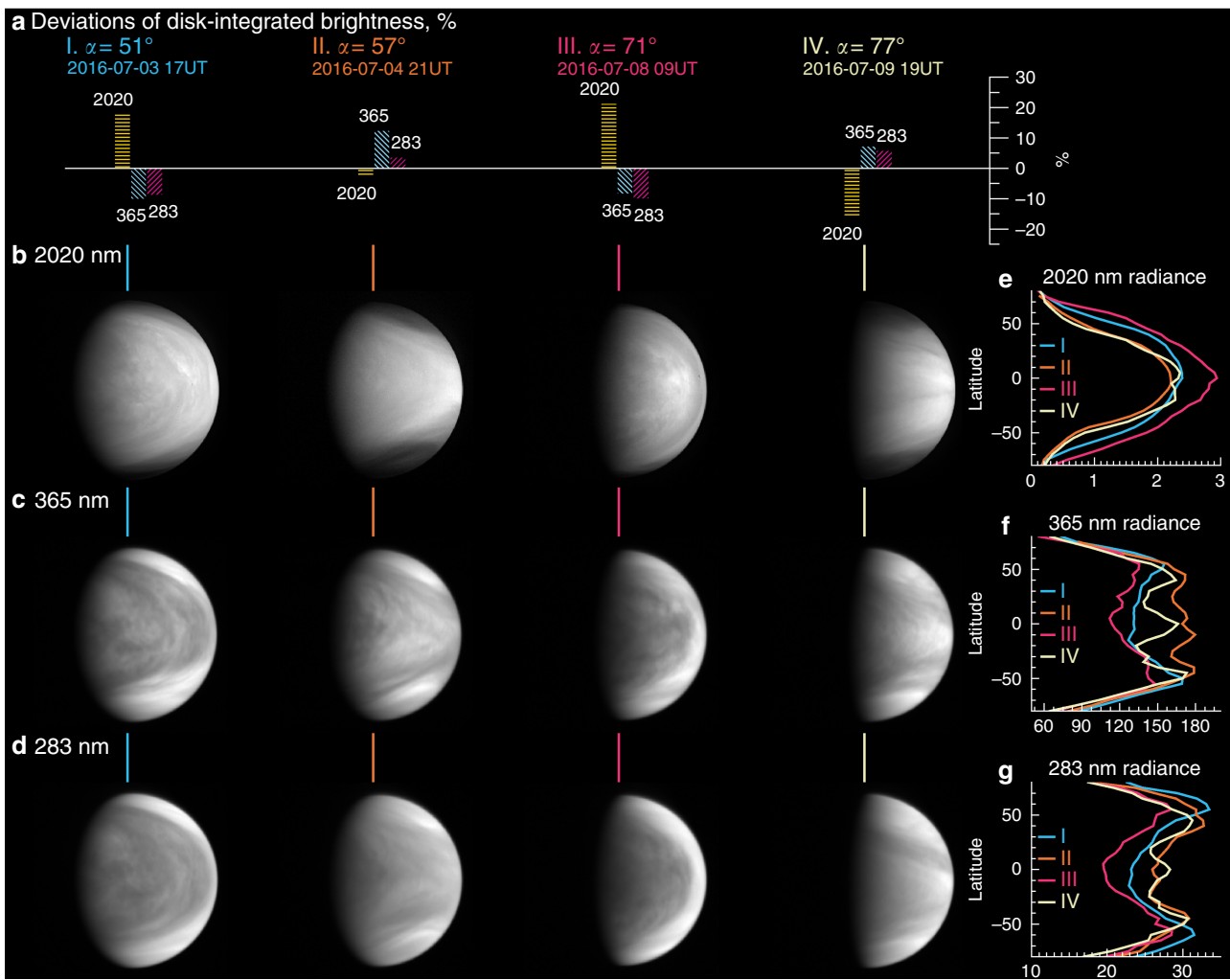

**Fig. 2 Variable cloud morphology during orbit 20 (2–10 July 2016). a** Deviation of disk-integrated brightness, %, from the mean phase curve (Fig. 1a). **b** Venus disk and brightness distributions at 2020 nm. All images show north at the top, white for high radiance, and black for zero radiance, and mostly the afternoon sides are visible. **c** Same but at 365 nm. **d** Same but at 283 nm. Roman numerals (I–IV) in **a** and vertical lines in (**b–d**) indicate three-wavelength images at specific solar phase angles (°) and times (YYYY-MM-DD HH in UTC). **e** 2020 nm latitudinal mean radiance [W m$^{-2}$ str$^{-1}$ μm$^{-1}$] at I–IV. **f** 365 nm latitudinal mean radiance at I–IV. **g** 283 nm latitudinal mean radiance at I–IV.

decreasing cloud top altitude[38]. These mid-to-high latitudinal variations are oscillated horizontally, in the latitudinal direction, by the Rossby wave[29], and thus $P_2$ becomes clear at 365 and 2020 nm.

Both the $P_1$- and $P_2$-period signals are recurrent features in Akatsuki's multi-year time-series of UV brightness measurements. The strength of each signal fluctuates over timescales of a few months (Fig. 4). This long-term evolution seems to follow the evolving viewing/illumination conditions introduced by the motion of Akatsuki and Venus on their orbits, also after removing the mean phase curve baseline. Even considering that the viewing/illumination geometry may affect the brightness deviations to some extent, the steep changes in strength of the $P_1$- and $P_2$-period signals for small changes in phase angle at 283 and 365 nm (Fig. 4) suggest that geometrical effects are not the primary cause of the periodograms' months-long fluctuations. It appears more credible that these fluctuations reflect real temporal variations in Venus' atmosphere. This alternating behavior between the $P_1$- and $P_2$-period signals has been described before in disk-resolved brightness investigations[31], and is thought to be connected with the processes that sustain the atmospheric superrotation.

## Discussion

The key features of the Venus periodogram for disk-integrated brightness are (i) it shows a single period ($P_1$) at 283 nm, but two non-fractional periods ($P_1$ and $P_2$, with $P_1/P_2$ and $P_2/P_1 \neq$ integer number) at 365 and 2020 nm; (ii) the brightness modulations in the UV and NIR are anti-correlated; (iii) the strengths of the $P_1$- and $P_2$-period signals exhibit long-term variations.

The above findings are relevant to the characterization of terrestrial exoplanets in reflected starlight with future space telescopes such as the Large UV/Optical/IR Surveyor (LUVOIR)[40] and Habitable Exoplanet Imaging Mission (HabEx)[41]. The logical next step here is to assess what could be learned from the above key features if they were identified in exoplanet data, and in particular how they could help discern whether the planet has an atmosphere and whether it is optically thin or thick. The exercise sets the basis for differentiating an exo-Venus from an exo-Earth before attempting to map out the planet's surface.

The detection of a single dominant period in a brightness periodogram does not by itself prove that there is an atmosphere. Indeed, geological inhomogeneities at the surfaces of atmosphere-less objects also produce brightness modulations[42]. Earth has a thin atmosphere and, for the same reason, the occurrence in its

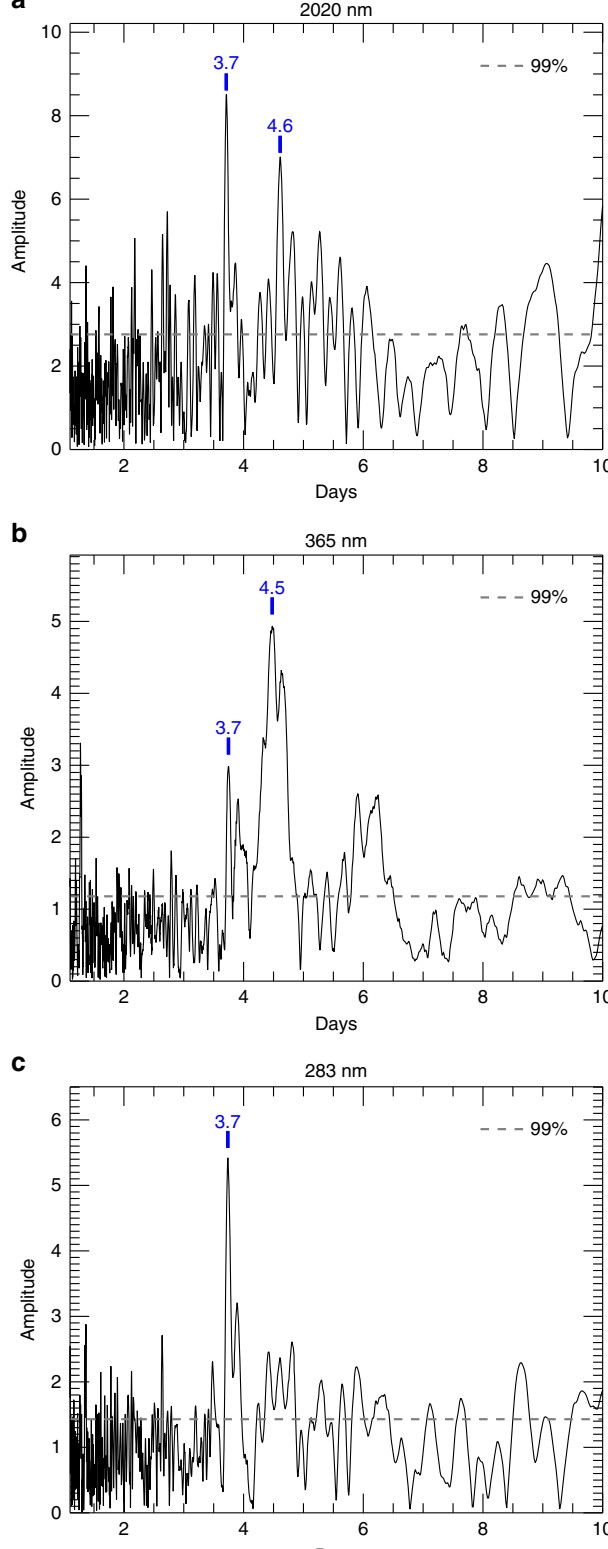

**Fig. 3 Periodograms of the deviations (%) in the disk-integrated brightness with respect to the mean phase curve in 2016.** (**a**) is at 2020 nm; (**b**) is at 365 nm; (**c**) is at 283 nm. The numbers in blue indicate the periods in Earth Days. 99% significance levels are also shown.

periodogram of a 1-day period (and additional fractional periods) cannot discriminate between a planet with or without an atmosphere. Further information might help if for example it provides evidence against a static surface albedo. This has been explored for Earth with the Transiting Exoplanet Survey Satellite (TESS)[43] broadband optical-NIR photometry[8], showing that aperiodic brightness fluctuations inconsistent with solid-body rotation hint at a dynamical atmosphere. Although not a terrestrial planet, it is also worth recalling that Neptune's periodogram, as determined with Kepler/K2 observations[44], exhibits a dominant peak at P ~ 17 h and smaller-amplitude peaks near 18 hours. The presence of discrete clouds altering Neptune's overall reflectance together with differential rotation of the background atmosphere induces the multiple periods. The small amplitude of the brightness modulations (<2%, peak-to-peak in the Kepler/K2 passband) and the close proximity of the periods, which will likely appear as a single period in exoplanet observations, will pose a severe challenge to distinguish Neptune's periodogram from that of an atmosphere-less object.

Unlike for Earth (or Neptune), the detection of two distinct non-fractional periods in Venus' periodogram offers insight into the atmosphere. Indeed, it is difficult to reconcile the occurrence of both periods with a surface origin of the associated brightness modulations. This implies that one or both of the periodic signals must originate in the atmosphere. In the first case, one could conceive that the atmosphere is optically thin and $P_1$ ($P_2$) is the planet's rotational period, and thus the observations are revealing a non-synchronous brightness modulation with a longer (shorter) period $P_2$ ($P_1$) on top of the surface's rotational modulation. In the second case, which is true for Venus, one could conceive that the atmosphere is optically thick and both the $P_1$- and $P_2$-period signals originate in the atmosphere. The bottom line is that key feature (i) alone reveals the existence of an atmosphere from a Venus-like periodogram. This is not a trivial conclusion as many of the planets that will be targeted by direct imaging will lack information as basic as their mass and radius that is essential to constraining their density and therefore their interior composition. The difficulty to infer the occurrence of an atmosphere with reflected-starlight measurements described above mirrors to some extent the difficulties encountered for close-in terrestrial exoplanets investigated with phase curves and currently available telescopes[45].

Key feature (ii) suggests also the existence of an atmosphere that through wavelength-dependent optical thickness effects might affect the brightness modulations with different signs at short and long wavelengths. Last, key feature (iii) requires an atmosphere that evolves over time, although it is not obvious if this sets a valuable constraint on its optical thickness. This latter key feature however demonstrates the importance of observing over a long temporal baseline to capture long-term variations in the planet's brightness.

In perspective, Venus offers a caution message against future attempts to relate the periods of brightness modulations to the solid-body rotation period of a planet, especially with a single wavelength or over temporal baselines that may not capture the evolution of the dominating planetary-scale waves in the planet's atmosphere. Multi-wavelength and long-baseline observations, as shown here, will be useful to discriminate between Earth- and Venus-like periodograms, and therefore between both planet types, although probably not unambiguously.

## Methods

**Image processing**. The number of images that we collected per Akatsuki orbit is shown in Supplementary Fig. 1. One orbit takes ~11 days. Orbit 1 started on 7 2015 Dec. The images at the three wavelengths (283, 365, and 2020 nm) were acquired with two cameras: UVI and IR2. IR2 stopped operating toward the end of

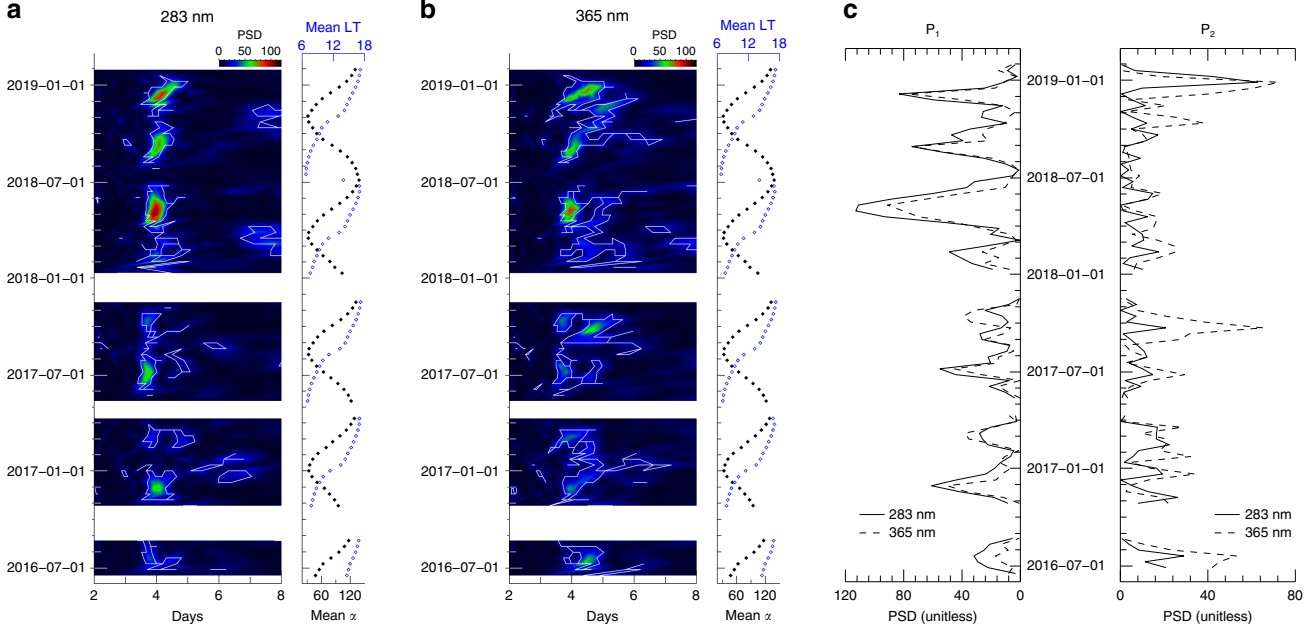

**Fig. 4 Long-term evolution of the periodograms at (a) 283 nm and (b) 365 nm.** The periodograms are calculated over 33-day windows (3 × Akatsuki's orbital period), which are sequentially shifted in steps of 11 days. Mean phase angle ($\alpha$, in degree) and mean local time (LT, in hours) are marked on the right-hand sides of (**a**) (**b**). The power spectral density (PSD) is shown in colors. The white solid line indicates the 99% significant level. **c** Comparison of PSD cross-sections at $P_1$ (3.9 days) and $P_2$ (4.6 days) (corresponding to the detected peaks in Supplementary Fig. 7) for both wavelengths.

2016. UVI has continued imaging Venus since the orbit insertion of the spacecraft. We analyzed images taken until 2019 January (orbit 105).

The radiance measured by UVI is corrected with the ground-measured flat-field, while the public data in DART (http://darts.isas.jaxa.jp/index.html.en) use the on-board diffuser flat-field. The flat conversion factor is publicly available through DART. We multiply the calibration correction factors ($\beta$) by the measured radiance: $\beta_{283} = 1.886$, $\beta_{365} = 1.525$, as described in Yamazaki et al.[15]. These are mean correction factors based on star observations between years 2010 and 2017, and are very close to the values reported in Yamazaki et al.[15]. The radiance measured by IR2 has a dependence on the sensor temperature. We correct for this dependence as described in Satoh et al.[46]:

$$I_{corr} = \begin{cases} I_{orig} / \left[ 1.0 - p_{59\,K} \left( \frac{T-T_0}{59-T_0} \right)^2 \right] & \text{for } T < T_0, \\ I_{orig} / \left[ 1.0 - p_{70\,K} \left( \frac{T-T_0}{70-T_0} \right)^2 \right] & \text{for } T \geq T_0, \end{cases} \quad (1)$$

where $I_{corr}$ is the corrected radiance, $I_{orig}$ is the measured radiance, $p_{59\,K} = 0.13$, $p_{70K} = 0.25$, $T_0 = 65.2$ K, and $T$ is the temperature of the sensor. We used images that had been treated by deconvolution of the point spread function (PSF), so the Venus image is sharper. The deconvolved IR2 images are available from the PI of the IR2 camera upon request.

We calculated the disk-integrated flux (units of W m$^{-2}$ $\mu$m$^{-1}$) from

$$F_{Venus}(\alpha, \lambda, t) = \sum_{r < r_o} I_{corr}(x, y) \times \Omega_{pix}, \quad (2)$$

where $(x, y)$ stands for pixel location on the image, $\Omega_{pix}$ is the pixel solid angle, $r$ is the distance of $(x, y)$ from the Venus disk center, and $r_o$ is the integration limit for the aperture photometry. We adopted $r_o = r_{Venus\ radius} + r_{PSF}$, where $r_{PSF}$ is the extent of the point spread function (7 pixels for UVI, 25 pixels for IR2). The PSF of the IR2 images is known to be wide[16], so we used the quoted value as a fine balance between signal and the required area of integration within the limited field of view (FOV, 12° × 12°). The center of Venus was found using the limb-fitting process[47]. We subtracted from the aperture photometry the background noise per pixel, estimated as the mean radiance over an outer ring around Venus between 40 and 70 pixels away from $r_{Venus\ radius}$ for UVI, and between 60 and 90 pixels for IR2. The solid angle of Venus, $\Omega_{Venus}$, is calculated as

$$\Omega_{Venus} = \pi \left( \sin^{-1} \left( \frac{R_{Venus\ radius}}{d_{V-obs}} \right) \right)^2, \quad (3)$$

where $R_{Venus\ radius}$ is the radius of Venus considering the cloud top altitude (=6052 +70 km), and $d_{V-obs}$ is the distance of the spacecraft from Venus in km.

Venus' brightness or phase-resolved albedo, as used in our work, is calculated through the following equation[48]:

$$A_{disk-int}(\alpha, \lambda, t) = \frac{\pi}{\Omega_{Venus}} \frac{d_{V-S}(t)^2 F_{Venus}(\alpha, \lambda, t)}{S_\odot(\lambda)}, \quad (4)$$

where $d_{V-S}(t)$ is the distance from Venus to the Sun [AU] at the time of observation $t$, $\Omega_{Venus}$ is the Venus solid angle (Eq. (3)), and $S_\odot(\lambda)$ is the solar irradiance at 1 AU (W m$^{-2}$ $\mu$m$^{-1}$) considering the transmittance functions of each filter. $S_\odot(\lambda)$ is taken from two sources. Near 365 and 2020 nm, we use the Smithsonian Astrophysical Observatory reference spectrum 2010[49]. Near 283 nm, we use the SORCE SIM Solar Spectral Irradiance (SSI) data (http://lasp.colorado.edu/home/sorce/data/ssi-data/ssi-data-file-summary/) after applying a 30-day running average. The $A_{disk-int}$ calculated in this study can be found in our Supplementary Data 1–3.

**Mean phase curves and periodicity analysis.** We estimated mean phase curves for the disk-integrated brightness at each wavelength $\overline{A_\lambda}(\alpha)$ in 2016 (Supplementary Fig. 2). At the UV wavelengths, $\alpha = 0°$–$20°$ was excluded to avoid the glory features[17–20]. The deviation of individual brightness measurements from the mean curve is calculated as

$$A_{devi, \lambda}(t) = \frac{A_{disk-int, \lambda}(\alpha, t) - \overline{A_\lambda}(\alpha)}{\overline{A_\lambda}(\alpha)} \times 100\,[\%]. \quad (5)$$

This helps remove the phase angle dependence from the brightness measurements, and allows us to focus on the time series for $A_{devi, \lambda(t)}$. Examples of deviations for orbit 20 (Fig. 1) are shown in Supplementary Fig. 3. The time series are subsequently used for the periodicity analysis (Fig. 3). To that end, we use the EFFECT software[50] with the algorithm in Deeming et al.[51]. The periodicity caused by the irregular data sampling, the so-called spectral window, has also been checked to look for overlapped peaks (none in our results). We repeated the same procedure for the entire 283 and 365 nm images (Supplementary Figs. 4–6). The full-time series at the three wavelengths are shown in Supplementary Fig. 6. We used those at UV for the periodicity analysis over years 2015–2019 (Supplementary Fig. 7).

Interestingly, the periodograms evolve over the multi-year span of our dataset. This evolution translates into relative variations in the strength of the $P_1$- and $P_2$-period peaks. The temporal evolution of the UV periodograms is shown in Fig. 4. For this particular figure, we used scargle.pro (http://astro.uni-tuebingen.de/software/idl/aitlib/timing/scargle.html; implementation from Press and Rybicki[52]) that is particularly efficient to process large data sets. We confirm the temporal variation in the signal strength from disk-integrated photometry for each of these periods, associated with Kelvin and Rossby planetary-scale waves, a finding consistent with what has been reported for disk-resolved photometry in previous studies[29–32]. For example, Imai et al.[31] (their Fig. 10) reported a transition from $P_1$ to $P_2$ at 365 nm from July to September 2017. This is also seen in Fig. 4b, where the peak shifts from $P_1$ to $P_2$ from September to October in 2017 at 365 nm. We also

note a clear shift in the identified periods from $P_1$ to $P_2$ in December 2018 at both 283 and 365 nm. The findings from our disk-integrated approach are also consistent with those from Nara et al.[32], who report a clear $P_1$ signal at low and middle latitudes in June 2018. Indeed, we can see in our Fig. 4a–b a very strong signal of $P_1$ in June 2018.

**The missing $P_2$ period at 283 nm in the context of Venus studies**. It is noteworthy that the periodogram at 283 nm (Fig. 3c) contains only evidence for the $P_1$ period. We propose that the $P_2$ period is missing at 283 nm because of the weaker absorption of the unknown absorber at this wavelength and the smooth latitudinal variation of $SO_2$[37]; both properties attenuate possible horizontal disturbances introduced by the Rossby wave. Additionally, increased Rayleigh scattering at 283 nm, especially in a slanted view, may suppress the specific signal of the mid-latitude Rossby wave. Also noteworthy, the $P_2$-period signal in the periodogram at 365 nm is significantly broader than the other peaks, a possible outcome of the stronger north–south asymmetry in brightness and wind speeds at this wavelength relative to 283 nm[53,54].

## Data availability
The UVI (level 3x products) and IR2 (level 3x geometry products) data that support the findings of this study are available in the JAXA archive website, http://darts.isas.jaxa.jp/ DARTS, with the identifiers https://doi.org/10.17597/ISAS.DARTS/VCO-00016[55] and https://doi.org/10.17597/ISAS.DARTS/VCO-00018[56], respectively. Deconvolved IR2 images are available from the PI of IR2 upon request, because the data are still experimental and subject to revision with an improved point spread function. All procedures of IR2 data improvement are documented and archived by the PI. Our disk-integrated brightness data are provided with this paper as Supplementary Data 1–3.

## Code availability
EFFECT software[50] is used for the periodicity analysis and scargle.pro (http://astro.uni-tuebingen.de/software/idl/aitlib/timing/scargle.html) is used to temporal evolution of the UV periodograms.

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

## Acknowledgements
The authors thank the Akatsuki team. Y.J.L. thanks Dr. Aleksandar Chaushev for discussion. Y.J.L. has received funding from EU Horizon 2020 MSCA-IF No. 841432.

## Author contributions
Y.J.L. and A.G.M. prepared the manuscript. A.G.M. conceived the main strategy, and Y.J.L. performed the data analysis and prepared figures. Y.J.L., A.G.M., T.I. interpreted the results. Y.J.L, and Y.M. worked on the UVI data quality maintenance. T.S. is the PI of IR2 and performed the calibration of IR2 data. A.Y. and S.W. maintained the UVI operation, and S.W. is the PI of UVI.

## Funding

## Competing interests
The authors declare no competing interests.
