## [Peer Review File · Nature Communications]

REVIEWER COMMENTS

Reviewer #1 (Remarks to the Author):

The paper reports some good and interesting work using recent data from the Akatsuki spacecraft currently orbiting Venus to address the timely question of the detection and characterisation of earth-like exoplanets. In particular, it synthesises whole-disc light curves of Venus at one infrared and two ultraviolet wavelengths in order to assess what might be learned from similar observations of an 'exoVenus' in orbit near the habitable zone of a distant star. It is shown that periodicities detected at multiple wavelengths can be used to help distinguish between a terrestrial planet with a relatively thin atmosphere, like Earth, where any regular modulation in brightness is mainly attributable to surface features, and a thick atmosphere like Venus where the fluctuations are caused by cloud variations due to large-scale planetary wave features.

The conclusion, rather unsurprisingly, is that such a distinction is possible but may not be unambiguous. The analysis is clearly presented and is likely to be of some interest to the community but to have real impact the research needs to reach the point where observations of light curves from actual candidate earth-like exoplanets can be presented and discussed in the light of the Venus analogue. This is a big ask, of course, but in the meantime the 'photometric pulse' of Venus has been extensively discussed and analysed elsewhere in the literature. This includes some excellent and very detailed work by some of the same authors using some of the same Akatsuki data, for instance Imai et al., *J. Geophys. Res.*, 124, Aug 2019, as well as other authors going back to the Pioneer Venus era more than 30 years ago (e.g. Del Genio and Rossow, *J. Atmos. Sci.* 47, 293, 1990).

Some suggestions for improving the presentation in the manuscript include the following points, some minor, others less so:

(i) Add more paragraph breaks, for instance at line 12, line 16, line 37 line 172.

(ii) In line 85, change 'be resulted' to 'result from'. There are other minor grammatical improvements that can be made elsewhere in the manuscript.

(iii) In lines 95-100 and elsewhere, clarify why the periodicities in the light curve identified in this work seem to differ significantly from the 'comparable' results in the references cited, including those which use the same Akatsuki data set.

(iv) The discussion of the results in lines 106-132 is confusing. It seems to be trying to extract from the disc-integrated data conclusions that are better obtained from disc-resolved maps and spectra and reported in the references cited and elsewhere. The point surely is to address the question of what could be inferred from photometric data from a Venus-like exoplanet, where there is no a priori spatially-resolved data. It would be better off this section was re-written in those terms.

(v) The general discussion in lines 133-160 belongs in the introduction, not in the conclusion.

(vi) The real conclusions section, beginning in line 161, is rather weak and too long. Anyone familiar with UV and IR observations of Earth and Venus would expect their periodograms to be quite different. Some redrafting, restricted to emphasising key conclusions and perhaps omitting the last two sentences (line 178-182), would be an improvement.

(vii) In Figure 1, the left and right panels should be scaled to match each other.

F W Taylor, University of Oxford

Saturday, 18 April 2020

Reviewer #2 (Remarks to the Author):

The manuscript by Y. J. Lee et al. presents an analysis of Venus photometric data taken by the Akatsuki spacecraft in which they probe the periodicity of albedo variations, linking them to planetary-scale atmospheric waves. The article is important for studies of Venus, but perhaps more so in the context of future observations of exoplanets. The authors convincingly demonstrate that it will be difficult to interpret photometric phase curves of exoplanets

in terms of whether the observed variability is due to atmospheric or surface features, but suggest ways in which this issue may be mitigated. I have only three comments for the authors, none of which are major.

1. My first comment is purely aesthetic and concerns the layout and readability of Figure 2. There is a lot going on in it, and it took me a while to fully digest all the information. Here are some suggestions to improve it:

- Please make it more obvious that each row of images corresponds to a particular wavelength. This is buried in the caption of the latitudinal profiles at the right, but should be a prominent label on the left hand side of the plot.
- Make the phase angles labels for each of the columns at the very top, rather than repeating them in the label for each sub panel.
- Reduce the font size of the "YYYYMMDD HHMMNSC" image captions, as this is relatively unimportant metadata.
- Use numbers for the rows and letters for the columns (or something analogous to this). Cross-referencing the letters across each of the panels is tedious for the reader. A row/column numbering scheme would help.

In Figure 1, please adjust the size of the right panel to match the size of the left panel -- the axes should be aligned across the two panels for better viewing, and the font sizes in the captions should also be adjusted to match.

2. The authors point to the fact that their Venus periodogram is distinct from Earth's periodogram -- in particular, the fact that it has two comparable dominant periods -- as a potential method to distinguish between optically thick and optically thin atmospheres. They claim that the two closely spaced periodogram peaks are difficult to reconcile with temporally static surface brightness variability. I think this argument is plausible, but I don't think it's the most robust way of differentiating between the two given the dataset. The DSCOVR-based power spectrum analysis of the Earth was based on monitoring of the Earth's light curve at full phase, in which case the dominant signals are the 1 day rotation period and its aliases (the 1/2, 1/3, and 1/4 figures quoted in the text are aliases -- they will show up for just about any continental distribution). But at phases closer to crescent, the power spectrum is much more sensitive to the exact spatial distribution of albedo on the surface, and peaks at various frequencies should appear. I agree that it's difficult to imagine two non-aliased nearby peaks arising from a static albedo distribution, but at low signal-to-noise (which will be the case for exoplanets) it will be very difficult to interpret these power spectra unambiguously.

A better way to distinguish between atmosphere and surface would be to try to isolate variability due to rotation and changes in phase, then look for additional variability on top of that, which must be temporal and thus (likely) associated with weather. This was done to some extent in

Luger et al. (2019, <https://arxiv.org/abs/1903.12182>), who analyzed TESS scattered light to infer the mean cloud cover map of the Earth. In section 5.4 they attempt to model the planet as a linear combination of a static component and a time-variable component, and find that the addition of the time-variable component somewhat improves the quality of the fit.

I appreciate the fact that the authors are doing this at some level by dividing out the phase dependence of the albedo. But perhaps a more robust way of testing whether or not the variability is due to static surface features or temporal atmospheric features is to find all the peaks in the power spectrum, and for each one, assume it corresponds to the solid body rotation period, then look at all measurements taken at the same rotational phase and illumination phase; any variability across those measurements above the level of the noise should be due to temporal changes. Inspecting the data in Supplementary Figure 6, it appears to my eye that there is in fact a signal at ~8 months that might correspond to the actual rotation period of the planet. So, in principle, if Venus were an exoplanet with unknown rotation period and with unknown surface/atmospheric properties, this procedure could be carried out to infer that we are likely not seeing the surface, but a temporally-variable cloud deck.

I don't think this sort of robust analysis is necessary for this paper to be published, but I would like the authors to at least discuss it. My main point is that there's a lot more information in the dataset beyond just the periodogram, and that the key to characterizing exoplanets

will lie in a careful modeling of the variability itself.

3. I am more familiar with the exoplanet literature on this, so the authors should feel free to defend their usage of the term, but I'm not sure the quantity the authors are reporting is technically an albedo. At full phase, integrating over the full disk does indeed yield the (geometric) albedo, but at any other phase, the quantity one gets is the geometric albedo times some unknown phase function that takes into account the phase dependence of the scattering but also the fact that the total illumination of the visible disk is decreased.

See, for instance, Figure 1 in Garcia-Munoz et al. (2014).

I'm concerned that what the authors are calling an "albedo" actually depends on the total amount of light incident on the portion of the Venusian surface visible to the spacecraft, which it shouldn't. In other words, it seems to me that the slope in Figure 1 is mostly due to the simple fact that there is more nightside in view, so the total flux visible from Venus is decreased. This shouldn't, however, affect the *albedo* of the planet.

Reviewer #1:

We thank Prof. Taylor for his valuable comments and his interest in our manuscript.

The conclusion, rather unsurprisingly, is that such a distinction is possible but may not be unambiguous. The analysis is clearly presented and is likely to be of some interest to the community but to have real impact the research needs to reach the point where observations of light curves from actual candidate earth-like exoplanets can be presented and discussed in the light of the Venus analogue. This is a big ask, of course, but in the meantime the 'photometric pulse' of Venus has been extensively discussed and analysed elsewhere in the literature. This includes some excellent and very detailed work by some of the same authors using some of the same Akatsuki data, for instance Imai et al., J. Geophys. Res., 124, Aug 2019, as well as other authors going back to the Pioneer Venus era more than 30 years ago (e.g. Del Genio and Rossow, J. Atmos. Sci. 47, 293, 1990).

(A) We agree that investigations like ours will have a stronger impact when the observations of Earth-like exoplanets become available. To reach that point, however, the technology must be developed and the key science cases must be laid out. Regarding the latter, our investigation shows the limitations of single-wavelength observations, and makes a strong case for multi-wavelength observations over long temporal baselines to infer whether an atmosphere exists on the planet, and ultimately to distinguish between an exo-Earth (thin atmosphere) and an exo-Venus (thick atmosphere). This key idea is emphasized further in the revised version of the manuscript.

As noted by the Reviewer, the brightness modulation of Venus (and the 4- and 5-days periods) has long been known. Our work differs from past works on this topic, some of which are cited in the manuscript, in a number of ways. The main difference is that we focus on the disk-integrated signal rather than on the signal over latitudinally- or longitudinally-resolved parts of the planet as our goal is to look at Venus as if it were an exoplanet. To the best of our knowledge, only Del Genio & Rossow (1982) have looked into the disk-integrated brightness modulation of Venus. Their study, however, which predates by more than a decade the discovery of exoplanets, focuses on the secular variation of Venus' reflectivity (their Fig. 8) rather than on how the 4-/5-day modulations could help distinguish between a planet with a thin/thick atmosphere. Also, the PVO images used by Del Genio & Rossow (1982, 1990) are specific to the wavelength of 365 nm, which prevents drawing conclusions on the value of multi-wavelength observations; it is unclear from Figure 8 of their 1982 paper whether the 5-day wave is present in the disk-integrated photometry.

It is fair to state that our work is the first one to set the disk-integrated brightness modulations of Venus in the context of exoplanet studies, and to explore the value of multi-wavelength observations over long temporal baselines. This demonstrates the synergy between solar system and exoplanet research.

Ultimately, we hope that our study will serve as a timely reminder to the exoplanet community that Venus (and foreseeably other cloudy planets) exhibits wave signatures in its atmosphere that might be potentially mistaken for surface features. In this respect, it's worth recalling that Ford et al. 2001

(<https://doi.org/10.1038/35091009>) states "This is in dramatic contrast to planets like Venus which would show almost no diurnal variability", and Palle et al. 2008 (<https://doi.org/10.1086/528677>) states "Some planets, such as Venus, are 100% cloud covered and show no significant photometric variability with time."

Indeed, our investigation shows that the reality is more complex than that.

(i) Add more paragraph breaks, for instance at line 12, line 16, line 37 line 172.

(A) We added more paragraph breaks, except in the Abstract.

(ii) In line 85, change 'be resulted' to 'result from'. There are other minor grammatical improvements that can be made elsewhere in the manuscript.

(A) Thanks for pointing this out. Upon reading the manuscript again, we have also identified a few other grammar issues.

(iii) In lines 95-100 and elsewhere, clarify why the periodicities in the light curve identified in this work seem to differ significantly from the ‘comparable’ results in the references cited, including those which use the same Akatsuki data set.

(A) The Table below (which we added to the S.I. as Table 1; see also Supplementary Fig. 8) summarizes the periods reported in a number of works. In general, our P1 and P2 periods fit within the typical values for the Kelvin and Rossby waves periods reported in the literature. The match is particularly good when, as done in other works, we focus on equatorial and middle latitudes separately. To clarify this important point, we have created a new section in the S.I. (plus Table 1 and Supplementary Fig. 8). The key message that our P1 and P2 periods are consistent with what has been reported in other works is now explicitly stated in the Main Text. It seems that not discriminating between equatorial and middle latitudes tends to push P2 from 4.9-5 days to 4.5-4.6 days. Whether this is caused in the process of averaging over latitude or over local time (for example, thermal tides impacting the background winds) requires further study. Such future study should compare the brightness with background winds to achieve a better interpretation.

Supplementary Table 1

Reference	Time of observation	Disk-integrated (wavelength)	Disk-resolved (365 nm)	
			Low latitude	Mid-latitudes
This study, Fig. 3 (283, 365, 2020 nm)	2016	3.7d (283 nm) 3.7 and 4.5-4.6d (365 and 2020 nm)	-	-
This study, Supplementary Fig. 7 (283, 365 nm)	2016-2018	3.9d (283 nm) 3.9 and 4.6d (365 nm)	-	-
This study, Supplementary Fig. 8 (365 nm)	2016-2018	-	0-20°N: 3.9d 0-20°S: 4.1d	40-60°N: 4.9d 40-60°S: 5.0d
Nara et al. 2020, Section 2.2.3	June 2018	-	Equator: 3.6±0.1d	Mid latitudes: 4.0±0.1d
Imai et al. 2019, Table 2 (365 nm)	90 days (18 Jun-16 Sep 2017)	-	0.75-3.75°S: Kelvin: 3.4-4.2d Rossby: 4.5-5.5d	44.25-47.25°S: Kelvin: --- Rossby: 4.5-5.5d
Del Genio et al. 1982, Summary, p.413 (365 nm)	66 days in 1979	-	Equator: 3.94±0.1d	Mid latitudes: 5.20±0.2d
Del Genio et al. 1990, Table 2 (365 nm)	1979-1985	-	Equator: 3.94-5.03d	Mid latitudes: 5.03-6.00d

(iv) The discussion of the results in lines 106-132 is confusing. It seems to be trying to extract from the disk-integrated data conclusions that are better obtained from disc-resolved maps and spectra and reported in the references cited and elsewhere. The point surely is to address the question of what could be inferred from photometric data from a Venus-like exoplanet, where there is no a priori spatially-resolved data. It would be better off this section was re-written in those terms.

(A) Our point was to give insight into the physics behind the P1 and P2 periods at all three wavelengths. We agree with the Reviewer that this level of detail may not be necessary/relevant for a paper that focuses on exo-Venus. We have truncated the corresponding discussion in the Main Text, and moved some of it to Methods.

(v) The general discussion in lines 133-160 belongs in the introduction, not in the conclusion.

(A) Thanks for the suggestion. We have moved most of it to the introduction, as proposed by the Reviewer.

(vi) The real conclusions section, beginning in line 161, is rather weak and too long. Anyone familiar with UV and IR observations of Earth and Venus would expect their periodograms to be quite different. Some redrafting, restricted to emphasising key conclusions and perhaps omitting the last two sentences (line 178-182), would be an improvement.

(A) As stated above, in the exoplanet community it is expected that the Venus periodogram must be flat because the planet is fully covered by clouds. This is obviously incorrect, as shown here. We agree with the Reviewer that the discussion could be streamlined, and we have tried to do so; the entire discussion section has been rewritten, emphasizing the key conclusions. We have also added to the discussion the case of Neptune's periodogram, which also exhibits multiple peaks. Correspondingly, also the Abstract has been rewritten to better reflect the main findings of our work.

(vii) In Figure 1, the left and right panels should be scaled to match each other.

(A) Thanks for the suggestion. We have made the changes.

Reviewer #2:

We thank Reviewer 2 for the constructive comments.

1. My first comment is purely aesthetic and concerns the layout and readability of Figure 2. There is a lot going on in it, and it took me a while to fully digest all the information. Here are some suggestions to improve it:

- Please make it more obvious that each row of images corresponds to a particular wavelength. This is buried in the caption of the latitudinal profiles at the right, but should be a prominent label on the left hand side of the plot.

- Make the phase angles labels for each of the columns at the very top, rather than repeating them in the label for each sub panel.

- Reduce the font size of the "YYYYMMDD HHMNSC" image captions, as this is relatively unimportant metadata.

- Use numbers for the rows and letters for the columns (or something analogous to this). Cross-referencing the letters across each of the panels is tedious for the reader. A row/column numbering scheme would help.

In Figure 1, please adjust the size of the right panel to match the size of the left panel -- the axes should be aligned across the two panels for better viewing, and the font sizes in the captions should also be adjusted to match.

(A) Thanks for these helpful suggestions. Figures 1 and 2 have been modified.

2. The authors point to the fact that their Venus periodogram is distinct from Earth's periodogram -- in particular, the fact that it has two comparable dominant periods -- as a potential method to distinguish between optically thick and optically thin atmospheres. They claim that the two closely spaced periodogram peaks are difficult to reconcile with temporally static surface brightness variability. I think this argument is plausible, but I don't think it's the most robust way of differentiating between the two given the dataset. The DSCOVR-based power spectrum analysis of the Earth was based on monitoring of the Earth's light curve at full phase, in which case the dominant signals are the 1 day rotation period and its aliases (the 1/2, 1/3, and 1/4 figures quoted in the text are aliases -- they will show up for just about any continental distribution).

But at phases closer to crescent, the power spectrum is much more sensitive to the exact spatial distribution of albedo on the surface, and peaks at various frequencies should appear. I agree that it's difficult to imagine two non-aliased nearby peaks arising from a static albedo distribution, but at low signal-to-noise (which will be the case for exoplanets) it will be very difficult to interpret these power spectra unambiguously.

A better way to distinguish between atmosphere and surface would be to try to isolate variability due to rotation and changes in phase, then look for additional variability on top of that, which must be temporal and thus (likely) associated with weather.

This was done to some extent in Luger et al. (2019, <https://arxiv.org/abs/1903.12182>), who analyzed TESS scattered light to infer the mean cloud cover map of the Earth. In section 5.4 they attempt to model the planet as a linear combination of a static component and a time-variable component, and find that the addition of the time-variable component somewhat improves the quality of the fit.

I appreciate the fact that the authors are doing this at some level by dividing out the phase dependence of the albedo. But perhaps a more robust way of testing whether or not the variability is due to static surface features or temporal atmospheric features is to find all the peaks in the power spectrum, and for each one, assume it corresponds to the solid body rotation period, then look at all measurements taken at the same

rotational phase and illumination phase; any variability across those measurements above the level of the noise should be due to temporal changes. Inspecting the data in Supplementary Figure 6, it appears to my eye that there is in fact a signal at ~8 months that might correspond to the actual rotation period of the planet. So, in principle, if Venus were an exoplanet with unknown rotation period and with unknown surface/atmospheric properties, this procedure could be carried out to infer that we are likely not seeing the surface, but a temporally-variable cloud deck.

I don't think this sort of robust analysis is necessary for this paper to be published, but I would like the authors to at least discuss it.

My main point is that there's a lot more information in the dataset beyond just the periodogram, and that the key to characterizing exoplanets will lie in a careful modeling of the variability itself.

(A) We agree with Reviewer #2 that temporal variability is a potentially valuable diagnostic to discriminate against a static surface albedo and therefore to infer the existence of an atmosphere. We have incorporated the discussion of temporal variability in our manuscript in a number of ways.

First, we have discussed more thoroughly than in our original manuscript that the Venus data reveal a months-long evolution of the P1- and P2-period signals (which seem to alternate), and moved Fig. 4 into the Main Text to emphasize this point. This long-term variability is evidence for an atmosphere, and demonstrates the importance of observing exoplanets over long temporal baselines. This should also be useful to the Venus community as the alternation of the P1 and P2 periods is possibly connected with the mechanisms that sustain the super-rotation of the atmosphere.

Second, we have noted that aperiodic brightness fluctuations (as shown for Earth with TESS photometry) will reveal an atmosphere. Related to this, we explicitly note that if e.g. P1 is assumed to be the solid body rotation period of the planet, then the planet's brightness will appear to fluctuate with the P2 period. And vice versa. Such fluctuation is not possible to reconcile with a static surface albedo.

Third, we include into the discussion of the periodograms the case of Neptune. Although not a terrestrial planet, it also exhibits multiple periods (although very closely centered near 18 hrs), which are caused by differential rotation of the atmosphere. The joint discussion of the periodograms for Earth, Neptune and Venus will hopefully give a broader perspective about the possibilities and limitations of using brightness time series in the future characterization of directly-imaged exoplanets.

*3. I am more familiar with the exoplanet literature on this, so the authors should feel free to defend their usage of the term, but I'm not sure the quantity the authors are reporting is technically an albedo. At full phase, integrating over the full disk does indeed yield the (geometric) albedo, but at any other phase, the quantity one gets is the geometric albedo times some unknown phase function that takes into account the phase dependence of the scattering but also the fact that the total illumination of the visible disk is decreased. See, for instance, Figure 1 in Garcia-Munoz et al. (2014). I'm concerned that what the authors are calling an "albedo" actually depends on the total amount of light incident on the portion of the Venusian surface visible to the spacecraft, which it shouldn't. In other words, it seems to me that the slope in Figure 1 is mostly due to the simple fact that there is more nightside in view, so the total flux visible from Venus is decreased. This shouldn't, however, affect the *albedo* of the planet.*

(A) The Reviewer is correct. The quantity that we are reporting is not a genuine albedo, and referring to it as an albedo may cause confusion. What we are reporting is traditionally described a geometric albedo \times phase law, i.e. $A_g \Phi(\alpha)$, with the convention $\Phi(\alpha=0)=1$. In most typical cases, $\Phi(\alpha)$ will decrease as α increases because the observer's view shifts towards the planet's nightside. However, this phase law may depend on atmospheric absorption, e.g. 2020 nm of this study due to CO₂ gaseous absorption, and specific haze atmospheric scattering, e.g. Titan's strong forward scattering. For these cases $\Phi(\alpha)$ increases at large phase angles.

To the best of our knowledge, there is no simple and universally accepted way of referring to $A_g \Phi(\alpha)$. To avoid confusion, we have modified the text to refer to $A_g \Phi(\alpha)$ as the planet's (size-normalized) brightness or phase-resolved albedo. Hopefully, the term "phase-resolved" will convey the idea that this is different from e.g. the geometric, Bond, or spherical albedos. We have added some text in the introduction to clarify the definition of the planet's brightness or phase-resolved albedo.

REVIEWERS' COMMENTS

Reviewer #1 (Remarks to the Author):

The authors have responded to the original review in a detailed and thoughtful way, and, given the use of important new data and the timely application to Venus and new world studies in the paper, I am now happy to see it published.

Reviewer #2 (Remarks to the Author):

I am pleased with the author's responses, and I thank them for addressing all my comments. I am happy to recommend the paper for publication.

Rodrigo Luger

Reviewer #1: *The authors have responded to the original review in a detailed and thoughtful way, and, given the use of important new data and the timely application to Venus and new world studies in the paper, I am now happy to see it published.*

We thank Prof. F. W. Taylor for his helpful review.

Reviewer #2: *I am pleased with the author's responses, and I thank them for addressing all my comments. I am happy to recommend the paper for publication.*

We thank Dr. Rodrigo Luger for his helpful review.